# A Case-Control Study on the Changes in Natural Killer Cell Activity following Administration of Polyvalent Mechanical Bacterial Lysate in Korean Adults with Recurrent Respiratory Tract Infection

**DOI:** 10.3390/jcm11113014

**Published:** 2022-05-26

**Authors:** Yun Kyong Lee, Ji-Hee Haam, Eunkyung Suh, Sung Hoon Cho, Young-Sang Kim

**Affiliations:** 1Chaum Life Center, CHA University, Seoul 06062, Korea; ykleefm@chamc.co.kr (Y.K.L.); hamjhi@chamc.co.kr (J.-H.H.); sherby@chamc.co.kr (E.S.); 2TS Bio, Seoul 08389, Korea; nk8275@naver.com; 3Department of Family Medicine, CHA Bundang Medical Center, CHA University, Seongnam 13496, Korea

**Keywords:** polyvalent mechanical bacterial lysate, natural killer cell activity, natural killer cell, recurrent respiratory tract infection, innate immunity

## Abstract

Polyvalent mechanical bacterial lysate (PMBL) is used for the treatment and prevention of recurrent respiratory tract infections. Although PMBL is an immunostimulant, it remains unknown whether treatment with PMBL influences natural killer cell activity (NKA). Hence, this case-control study compared the changes in IFN-γ levels (surrogate index for NKA) following PMBL treatment or time passing between the PMBL-treated group and controls. The treatment group included adults who had a PMBL prescription for three months against recurrent respiratory tract infection from an outpatient clinic, while the control group had healthy adults visiting the health promotion center for periodic health check-ups. The control group (N = 506) showed no change in IFN-γ levels, while the treatment group (N = 301) showed a significant increase in mean from 462.8 to 749.3 pg/mL after PMBL treatment. In the subgroup with IFN-γ <500 pg/mL, IFN-γ levels significantly increased in both groups. However, the change in the treatment group (287 ± 822 pg/mL) was greater than that in the control group (58 ± 809 pg/mL), and the interaction between the visit and case/control was significant (*p* = 0.030) in a generalized estimating equation model. In conclusion, NKA increased in the subjects with recurrent respiratory tract infections with PMBL treatment.

## 1. Introduction

Natural killer (NK) cells play crucial roles in innate immunity; they recognize and eliminate virus-infected and neoplastic cells [1]. Moreover, they regulate various immune cells, such as macrophages and dendritic cells [2]. Consequently, NK cells have roles in pulmonary infection and inflammation [3]. A previous study reported a reduction in NK cell activity (NKA) in children with recurrent respiratory tract infections [4]. Patients with severe COVID-19 have also shown a reduction in NK cell count, function, and cytolytic activity [5,6].

Polyvalent mechanical bacterial lysate (PMBL) is prepared by the mechanical lysis of various types of bacteria [7]. The lysate exerts immunomodulatory effects against the bacterial strains composing it, usually those responsible for respiratory infections [8,9]. The immunomodulation of PMBL affects immune cells, such as T, B, and NK cells [10,11,12]. The non-specific defense mechanisms of PMBL may lead to an increase in secretory immunoglobulin A in mucous membranes, phagocytic activity, and IFN-γ production [13].

It remains unknown whether the efficacy of PMBL is associated with a change in NK cell function. Studies have shown activation in NKA in both animals [14,15] and humans [16] after exposure to an influenza virus. Mimicking exposure to pathogens, PMBL administration may influence NKA and construct subsequent immunity to recurrent respiratory tract infections. Likewise, a small study reported that the absolute number of NK cells significantly increases in PMBL-treated chronic obstructive pulmonary disease (COPD) patients [10]. Recently, a novel assay has been developed to measure NKA in the serum of ex vivo stimulated whole blood to detect secreted IFN-γ from NK cells [17]. The simplicity of this method enabled the commercial use of NKA in large-scale studies [18,19]. However, it has not been investigated whether PMBL administration increases NKA when measured using this novel assay in patients with recurrent respiratory tract infections. In this case-control study, we compared the changes in NKA between patients who underwent PMBL treatment and healthy controls.

## 2. Subjects and Methods

### 2.1. Study Subjects

This case-control study was conducted based on the data between 2016 and 2020 from the Chaum Life Center (Seoul, Korea). The treatment group included individuals treated in outpatient clinics. The inclusion criteria were as follows: (1) patients with recurrent respiratory tract infection under PMBL treatment for three months following the recommended protocol of 10 days of sublingual administration and 20 days of drug-free interval, and (2) those who have undergone NKA assays before and after PMBL treatment. A total of 311 patients were included in the treatment group according to these criteria. The control subjects were from the health promotion center belonging to the Chaum Life Center. The inclusion criteria for the control group were as follows: (1) people who have undergone NKA assays in the health promotion center periodically at 1-year intervals, and (2) those who were not treated with PMBL between the visits. The control group consisted of 549 people according to these criteria. The endpoint was the change in IFN-γ levels, a surrogate index for NKA, from baseline to follow-up.

Those who had a history of malignancies or suspicious findings for any cancer were excluded (N = 16). We also excluded those with autoimmune disorders (N = 4), active allergic disease under treatment (N = 5), acute or chronic infectious disease (N = 7), and recent medication with antibiotics, immunosuppressants, or herbal medicines (N = 21). Finally, 301 patients and 506 controls were included in the analyses for the study (Figure 1).

The study was conducted following the guidelines of Helsinki’s Declaration and was approved by the Institutional Review Board of CHA Bundang Medical Center (2021-04-034).

### 2.2. PMBL

Ismigen^®^ (50 mg; Schumitt Health Korea LLC, Hanam, Korea) was prescribed to the treatment group as a PMBL. Ismigen^®^ was prepared from *Staphylococcus aureus*, *Streptococcus pyogenes*, *Streptococcus viridans*, *Klebsiella ozaenae*, *Haemophilus influenzae* serotype B, *Moraxella catarrhalis,* and *Streptococcus pneumoniae* by mechanical lysis. The protocol involved the sublingual administration of one tablet of Ismigen^®^ per day for 10 days, followed by a drug-free interval of 20 days for three consecutive months.

### 2.3. Medical History, Measurements, and Blood Sampling

The medical histories of the subjects were obtained. Height and weight were measured in a standing position without shoes and were recorded to the first decimal point in centimeters and kilograms, respectively. The body mass index (BMI) was defined as body weight in kilograms divided by the height squared in meters.

The blood samples were obtained in the morning after the patient had fasted overnight for at least 8 h and were derived from the antecubital area. The serum samples were stored at 4 °C and were analyzed within a day of sampling. The neutrophil and lymphocyte counts were estimated as a proportion of the number of white blood cells (WBC). The neutrophil-lymphocyte ratio (NLR) was defined as the neutrophil count divided by the lymphocyte count.

### 2.4. IFN-γ Measurement for NKA

IFN-γ was measured with a recently developed blood test (NK Vue^®^ Kit, NKMAX, Seongnam, Korea). Using a direct vacutainer system, a 1 mL sample of whole blood was directly transferred into a specific tube containing a patented stimulatory cytokine (Promoca^®^, NKMAX, Seongnam, Korea) for the tests. The collection tube was gently and repeatedly mixed, and within 30 min of collection, the tube was incubated for 20–24 h in a 37 °C chamber, following the manufacturer’s instructions. During the incubation period, the stimulatory cytokine caused the secretion of IFN-γ into the plasma; this secretion of IFN-γ predominantly occured through NK cells rather than other innate or adaptive immune cells [16,20,21]. Following incubation, the supernatant was obtained and centrifuged at 3000× *g* for 3 min. The supernatant was then loaded onto enzyme-linked immunosorbent assay (ELISA) plates. The IFN-γ levels were measured using a designed ELISA, reported by the units of pg/mL. This study used the reference ranges given by the test-kit manufacturer; IFN-γ ≥ 500 pg/mL as normal and IFN-γ < 500 pg/mL as low [19,22].

### 2.5. Statistical Analysis

The general features were expressed as mean ± SD or number (proportion). The differences between the treatment and control groups were compared using an independent sample *t*-test and a chi-square test. The description and comparison between the treatment and control groups were repeated in the subgroup with low and normal NKA (<500 pg/mL IFN-γ).

To confirm the factors influencing NKA levels, linear regression models with a stepwise selection method were used on all the subjects at baseline.

The changes in NKA values from baseline to the follow-up were compared using paired *t*-tests in the treatment and control groups. To compare the changes in NKA between the treatment and control groups, generalized estimating equation (GEE) models were formulated. In the GEE models, the difference between the treatment and control groups, between the baseline and follow-up visit, and their interactions were considered independent variables. Other factors, such as age, sex, medical history, and baseline laboratory data, were regarded as covariates, and significant variables were selected and included in the models. These statistical methods were repeatedly applied in the subgroups.

All of the statistical analyses were conducted using the SPSS statistical package, version 26 (IBM, Armonk, NY, USA). The results with *p* ≤ 0.05 were considered statistically significant.

## 3. Results

### 3.1. Baseline Characteristics of the Subjects

The baseline features of the subjects are indicated in Table 1. The mean age was 55.0 ± 13.6 years and 53.3 ± 10.6 years in the treatment and control groups, respectively. The WBC and neutrophil counts were higher in the treatment group than in the control group. IFN-γ levels were higher in the control group than in the treatment group. In the subgroup with low NKA, age and IFN-γ levels were not significantly different between the treatment and control groups.

### 3.2. Factors That Influence Baseline IFN-γ Levels in the Whole Subjects

Various factors such as age, sex, history, BMI, glucose, and complete blood count (CBC) parameters were considered covariates in the linear regression models (Figure 2 and Table 2). The stepwise method extracted significant variables, namely, age and CBC parameters. In Model 1, the IFN-γ levels were positively associated with age and lymphocyte count and negatively with the neutrophil count. In Model 2, the NLR was negatively associated with the IFN-γ levels. 

### 3.3. Changes in NKA from Baseline to Follow-Up

In the treatment group, the mean IFN-γ levels significantly increased from 462.8 to 749.3 pg/mL (Table 3 and Figure 3). In the control group, the IFN-γ levels did not show a significant change (961.7–1019.9 pg/mL).

To compare the changes in IFN-γ levels between the treatment and control groups, we created a GEE model that included variables of visit, case/control, their interaction, and covariates (Table 4). In the GEE model, the changes in IFN-γ levels were different between the treatment and control groups (*p* = 0.003), and the interaction between visit and case/control was also significant (*p* < 0.001). This is in agreement with the results of the linear regression model, where age and NLR were also shown to be significant factors.

In the subgroup of low NKA, both the treatment and control groups showed a significant increase in IFN-γ levels from baseline to follow-up (Table 3 and Figure 3). The mean changed from 213.1 pg/mL to 596.6 pg/mL and from 234.5 pg/mL to 517.3 pg/mL in the treatment and control groups, respectively. According to the GEE models of the subgroup with low NKA (Table 4), although the difference in IFN-γ level changes between the treatment and control groups was not significant, the interaction between the visit and case/control was significant (*p* = 0.030). In this model, the other covariates were not significant.

## 4. Discussion

The case-control study showed that the IFN-γ levels increased in the treatment group but not in the control group. In the subgroup with low NKA (<500 pg/mL IFN-γ), the mean increment was significantly higher in the treatment group than in the control group. Compared to the control group, the PMBL treatment group with recurrent respiratory tract infection showed a greater increase in IFN-γ levels, a surrogate index for NKA.

Since PMBL is prepared from various pathogens, PMBL is involved in acquired immunity [10,12]. Additionally, studies have shown that PMBL is also involved in innate immunity [10,12,23,24,25,26]. Accordingly, PMBL treatment resulted in an increase in NK cell counts. The absolute number of NK cells significantly increased by 56% in 28 COPD patients treated with PMBL [10]. A randomized controlled trial has shown an increase in NK cell count in 21 asthmatic children compared to the control group after PMBL treatment [27]. Our study showed a significant increase in IFN-γ levels in 301 patients treated with PMBL. This is the first report on the change in IFN-γ levels after PMBL treatment measured using the novel assay.

Infections may induce NKA augmentation. Exposure to influenza virus-infected cells has been shown to increase NKA [16]. Cytokine interactions during viral infections regulate NK cell activation and highlight variations in NK cell function [28]. Some vaccines contain weakened or inactive parts of a particular organism that triggers an immune response within the body [29]. Hence, vaccines may mimic the infection process, triggering an immune response, including NK cell activation. NK cell response in humans has been shown to increase after vaccination with influenza and Ebola virus vaccines [30,31]. PMBL is an oral immunostimulant consisting of standardized bacterial lysates obtained by the lysis of different strains of Gram-positive and Gram-negative pathogens that cause respiratory tract infections [12]. PMBL is known to improve the protective antibody response against respiratory viruses through Toll-like receptor 4 [32]. PMBL may share the immune response, including NK cell activation, with other vaccines.

Our previous report has shown that NLR is strongly associated with IFN-γ levels [19]. In this study, NLR also had a significant relationship with IFN-γ levels at baseline and follow-up. The association between NLR and IFN-γ levels is consistent with the previous findings that low IFN-γ levels may suggest inflammation. In the models for the subjects with low baseline NKA (<500 pg/mL IFN-γ), the range of IFN-γ levels was narrow, and NLR was not a significant factor. The previous study also showed an increase in IFN-γ levels in subjects with a low baseline NKA [18]. In the treatment group of the present study, it was unclear whether the increase in IFN-γ levels after treatment was caused by the PMBL or by time passing. In the subgroup with low NKA, the change in IFN-γ levels at baseline and follow-up was different, and the interaction between group and time was significant. These findings suggest that PMBL treatment may increase IFN-γ levels more than time.

Our study has several limitations. First, since the study employed a case-control design, the findings are not confirmative. Moreover, the mechanism of the effect of PMBL remained unclear. Further studies including RCTs are required to find the effect of PMBL on NKA, and a basic investigation to unfold the molecular mechanism should be undertaken. Second, we do not have enough data on the long-term prevention of respiratory infection and adverse events. Further clinical information should be warranted in the next studies. Third, the control group did not include subjects with recurrent respiratory tract infections. The control group included individuals who had routine health check-ups and NKA examinations repeated in a year, which is different from the treatment group with three-month-interval assays. However, in the subjects with low NKA, the baseline IFN-γ levels were not significantly different between the groups. The analysis for the subgroup attenuated the heterogeneity between the groups. Fourth, NKA was measured using only IFN-γ assays. NKA is routinely measured by perforin and granzyme B production, degranulation, or in a direct cytotoxicity assay against K562 cells [33]. Although NK cell count was not measured in this study, also the methods used are more appropriate for the analyses of a large number of samples. Fifth, contrary to the previous study [18], age was positively correlated with IFN-γ levels in our analysis. As a case-control study, the treatment group was selected only among those fit for the inclusion criteria. The young adults included in the treatment group had relatively lower IFN-γ levels than the older subjects. It is not proper, based on our findings, to express that older people have higher IFN-γ levels than younger ones.

## 5. Conclusions

PMBL treatment may increase IFN-γ levels, a surrogate index for NKA, in subjects with recurrent respiratory tract infections. It may suggest the influence of PMBL on innate immunity in the community setting. Further RCTs are required to find the effect of PMBL on the immunologic changes in the homogenous population.

## Figures and Tables

**Figure 1 jcm-11-03014-f001:**
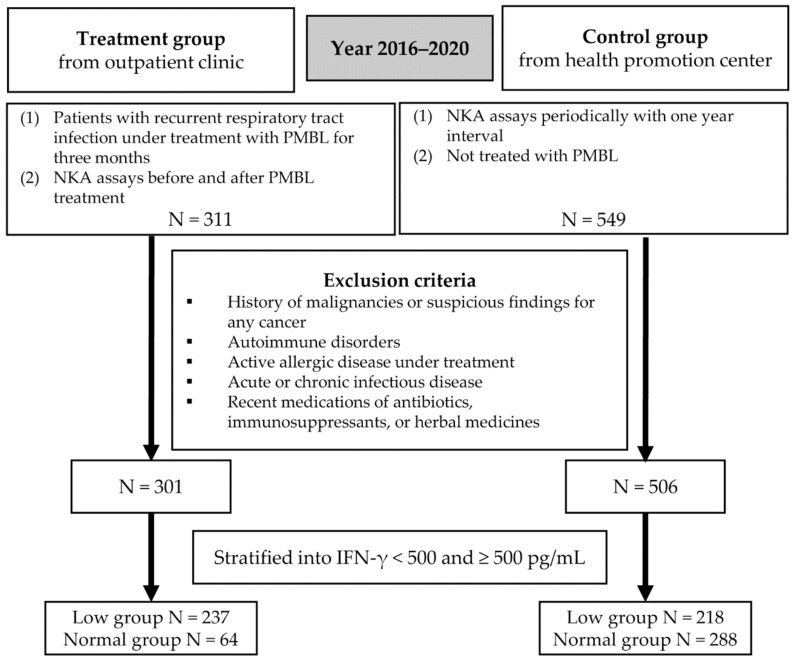
The flowchart of the study. PMBL, polyvalent mechanical bacterial lysate; NKA, natural killer cell activity.

**Figure 2 jcm-11-03014-f002:**
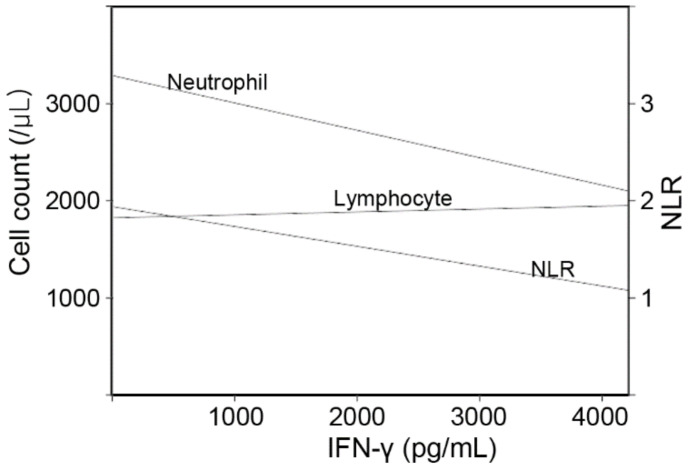
The linear trend lines of neutrophil, lymphocyte counts and their ratio following IFN-γ levels. The trend lines were drawn using linear regression analyses.

**Figure 3 jcm-11-03014-f003:**
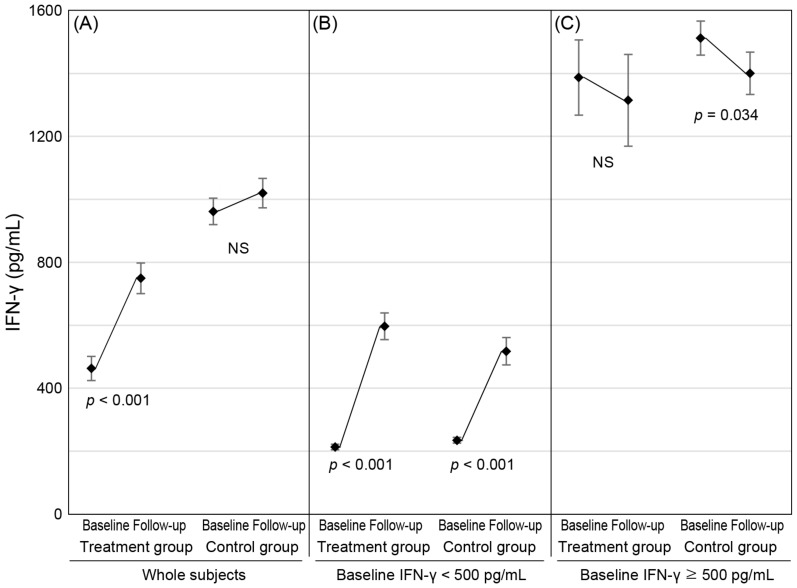
Changes in IFN-γ levels from baseline to follow-up in the treatment and control groups. The markers indicate mean and error bars show sem. (**A**) Analyses of the whole subjects show a significant increase in IFN-γ levels only in the treatment group. (**B**) In the subgroup with low baseline IFN-γ levels, IFN-γ levels significantly increased in both groups. (**C**) In the subgroup with normal baseline IFN-γ levels, IFN-γ levels significantly decreased in the control group. *p* values were calculated using paired *t*-test.

**Table 1 jcm-11-03014-t001:** The baseline characteristics of the patient and control groups.

	Whole Subjects			Baseline IFN-γ < 500 pg/mL			Baseline IFN-γ ≥ 500 pg/mL		
	Treatment Group	Control Group	*p*	Treatment Group	Control Group	*p*	Treatment Group	Control Group	*p*
	(N = 301)	(N = 506)		(N = 237)	(N = 218)		(N = 64)	(N = 288)	
Age (years)	55.0 ± 13.6	53.3 ± 10.6	0.052	53.7 ± 13.6	52.4 ± 10.2	0.248	60.0 ± 12.3	54.1 ± 10.8	<0.001
Sex (men)	118 (39.2%)	255 (50.4%)	0.003	89 (37.6%)	112 (51.4%)	0.004	29 (45.3%)	143 (49.7%)	0.624
Hypertension	40 (13.3%)	95 (18.8%)	0.055	29 (12.2%)	43 (19.7%)	0.040	11 (17.2%)	52 (18.1%)	1.000
Dyslipidemia	73 (24.3%)	121 (23.9%)	0.981	57 (24.1%)	44 (20.2%)	0.380	16 (25.0%)	77 (26.7%)	0.898
Diabetes	22 (7.3%)	25 (4.9%)	0.217	17 (7.2%)	9 (4.1%)	0.232	5 (7.8%)	16 (5.6%)	0.691
Body mass index (kg/m^2^)	22.7 ± 3.1	23.5 ± 3.3	0.002	22.7 ± 3.2	23.5 ± 3.4	0.012	22.9 ± 2.7	23.4 ± 3.2	0.183
Glucose (mg/dL)	92.2 ± 15.4	91.5 ± 19.3	0.570	91.7 ± 15.8	91.1 ± 20.4	0.697	93.9 ± 13.8	91.8 ± 18.4	0.397
WBC (/µL)	5621 ± 1464	5386 ± 1444	0.027	5627 ± 1433	5676 ± 1587	0.728	5601 ± 1587	5167 ± 1286	0.020
Neutrophil (/µL)	3205 ± 1173	2989 ± 1092	0.010	3222 ± 1166	3290 ± 1219	0.540	3145 ± 1206	2761 ± 923	0.005
Proportion (%)	56.1 ± 8.4	54.8 ± 8.7	0.027	56.4 ± 8.5	57.3 ± 8.9	0.291	55.2 ± 8.0	52.9 ± 8.1	0.039
Lymphocyte (/µL)	1845 ± 533	1849 ± 563	0.919	1837 ± 528	1834 ± 610	0.953	1875 ± 557	1861 ± 525	0.843
Proportion (%)	33.6 ± 8.1	35.0 ± 8.0	0.018	33.4 ± 8.1	32.9 ± 8.1	0.504	34.3 ± 8.2	36.6 ± 7.6	0.039
NLR	1.85 ± 0.81	1.74 ± 0.84	0.055	1.87 ± 0.80	1.97 ± 1.04	0.242	1.79 ± 0.86	1.56 ± 0.60	0.043
NKA (pg/mL)	463 ± 662	962 ± 942	<0.001	213 ± 139	235 ± 135	0.096	1387 ± 955	1512 ± 917	0.329

NKA, natural killer cell activity; BMI, body mass index; WBC, white blood cells; NLR, neutrophil-lymphocyte ratio.

**Table 2 jcm-11-03014-t002:** Factors that influence natural killer cell activity in the whole subjects at baseline derived from the linear regression analyses.

	Beta (SE)	*p*
Model 1		
Age	0.085 (0.034)	0.013
Neutrophil	−0.234 (0.035)	<0.001
Lymphocyte	0.102 (0.035)	0.004
Model 2		
Age	0.094 (0.034)	0.006
NLR	−0.215 (0.034)	<0.001

The variables in the regression model were determined through the stepwise selection method. NLR, neutrophil–lymphocyte ratio.

**Table 3 jcm-11-03014-t003:** Comparison of changes in IFN-γ levels from baseline to follow-up between the treatment and control groups.

	Treatment Group	Control Group	*p*
Whole subjects	287 ± 822	58 ± 809	<0.001
Low NKA	384 ± 638	283 ± 615	0.087
Normal NKA	−72 ± 1237	−112 ± 893	0.809

Low NKA is defined as IFN-γ levels < 500 pg/mL. NKA, natural killer cell activity.

**Table 4 jcm-11-03014-t004:** The unstandardized coefficients of the variables in the generalized estimating equation analyses.

	Whole Subjects			Baseline IFN-γ < 500 pg/mL			Baseline IFN-γ ≥ 500 pg/mL		
	Coefficients	SE	*p*	Coefficients	SE	*p*	Coefficients	SE	*p*
Baseline to follow-up	0.533	0.084	<0.001	1.030	0.075	<0.001	−0.024	0.124	0.848
Treatment group (vs. control)	−0.245	0.083	0.003	0.137	0.111	0.218	0.008	0.125	0.948
Interaction between visit and group			<0.001			0.030			0.618
Age (years)	0.010	0.003	0.002	0.001	0.003	0.738	−0.005	0.003	0.128
NLR	−0.247	0.043	<0.001	−0.049	0.041	0.230	0.232	0.051	<0.001

NLR, neutrophil-lymphocyte ratio.

## Data Availability

Not applicable.

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
