# Peer review of "A Case-Control Study on the Changes in Natural Killer Cell Activity following Administration of Polyvalent Mechanical Bacterial Lysate in Korean Adults with Recurrent Respiratory Tract Infection"

_jcm, 2022, doi:10.3390/jcm11113014_

Round 1
Reviewer 1 Report
The authors present an interesting manuscript about the activity of NK cells in a patient population with recurrent respiratory tract infections compared to healthy controls. The use of the English language was decent up until the discussion section, which could benefit from considerable revisions, paying particular attention to syntax and grammar. One of the downfalls of this study, as mentioned by the authors is that the intervention group is an unhealthy group receiving a treatment that the authors compare to a healthy group. Perhaps one way to circumvent this issue is to compare each patient to themselves.
One question that lingers is the capability of the NKA assay, which may be inappropriately used outside of its limit of detection (i.e. <500 pg/mL interferon-gamma). There are several statistical p values reported that do not appear to be powered by the study. A formal power analysis would be advisable.
On that note, specifically outlying the primary and secondary measurements of this study would help to clarify the main focus of the authors. As it stands, it appears as though there were dozens of characteristics that were evaluated using a statistical test with alpha set at 0.05, meaning that for every twenty comparisons there will be one that is inappropriately marked as significant.
Another concern is the groups were separated into "whole" and "<500 pg/mL" but there is no ≥500 pg/mL group, which is perplexing. Why arbitrarily set <500 pg/mL for a group? Is it solely because the developers of the NKA assay said <500 pg/mL is low?
The authors' choice of measures of central tendency are also confusing. For example, the NKA assay data are displayed in both parametric mean +/- SD and non-parametric median +/- IQR, yet the statistical tests of significance are performed only on non-parametric values.
In the end, there are several concerns in this manuscript. However, the authors have an interesting concept that would benefit the scientific community by being published.

Author Response
The authors present an interesting manuscript about the activity of NK cells in a patient population with recurrent respiratory tract infections compared to healthy controls. The use of the English language was decent up until the discussion section, which could benefit from considerable revisions, paying particular attention to syntax and grammar. One of the downfalls of this study, as mentioned by the authors is that the intervention group is an unhealthy group receiving a treatment that the authors compare to a healthy group. Perhaps one way to circumvent this issue is to compare each patient to themselves.
Response: Thank you for the interest in our study. As the reviewer commented, a more comprehensive design would be the comparison in an unhealthy population with and without a specific medication. However, since this study is based on the medical records, a part of the patients has received the prescription. We are aware of this limitation, and it was described in the limitation section. If possible, clinical trials may help in discovering the effect of PMBL on NK cell activity.
One question that lingers is the capability of the NKA assay, which may be inappropriately used outside of its limit of detection (i.e. <500 pg/mL interferon-gamma). There are several statistical p values reported that do not appear to be powered by the study. A formal power analysis would be advisable.
Response: As the reviewer commented, we clarify the primary endpoint. The main comparison of NKA changes between two groups follows alpha (0.05). Additionally, only one subgroup with NKA < 500 pg/mL IFN-gamma (this term is also revised in the whole text) was re-evaluated. Then, the comparison of NKA change means between two groups gives rise to power of 97%.
On that note, specifically outlying the primary and secondary measurements of this study would help to clarify the main focus of the authors. As it stands, it appears as though there were dozens of characteristics that were evaluated using a statistical test with alpha set at 0.05, meaning that for every twenty comparisons there will be one that is inappropriately marked as significant.
Response: As described above, the primary endpoint is clarified as the changes in NKA according to the groups.
Another concern is the groups were separated into "whole" and "<500 pg/mL" but there is no ≥500 pg/mL group, which is perplexing. Why arbitrarily set <500 pg/mL for a group? Is it solely because the developers of the NKA assay said <500 pg/mL is low?
Response: The whole group includes the study subjects regardless of NKA values. In the whole subjects, baseline NKA was substantially different between treatment and control group. Our published study (Lee et al. 2022) showed that, in the subjects with NKA <500 pg/mL, NKA increases significantly following passed time. Hence, we used the cutoff value of 500 pg/mL. Concomitantly, the manufacturer also suggested this value as reference, too. We additionally described the parameters in the subjects ≥500 pg/mL.
The authors' choice of measures of central tendency are also confusing. For example, the NKA assay data are displayed in both parametric mean +/- SD and non-parametric median +/- IQR, yet the statistical tests of significance are performed only on non-parametric values.
Response: Although the distribution in NKA is right-skewed, the number of subjects is large enough to select parametric analyses. We decided to show parametric parameters only.
In the end, there are several concerns in this manuscript. However, the authors have an interesting concept that would benefit the scientific community by being published.
Response: Thank you.
How can one compare healthy and unhealthy+treated groups and know whether the effect measured is due to the treatment, the health status, or a combination of the two?
Response: As mentioned before, this grouping method is a limitation. Although we thought that a combination of treatment and time factor may affect NKA values, the expected change in the unhealthy group may not be completely identified with the time-dependent change in healthy controls.
It is unclear how NK cell activity can be measured in units of pg/mL. Why not measure NK cell count?
Response: We added the measurement process in the method section. As the reviewer commented, unit of pg/mL is applied to IFN-γ. In addition, since this study is based on medical records, NK cell count was not measured. This point is also a limitation, and we added this in the limitation part of Discussion.
I appreciate the cautious tone of this conclusion.
Response: The sentence was corrected as follows: NKA increased in the subjects with recurrent respiratory tract infections with PMBL treatment.
What is the specific immunostimulant? Do the authors mean a specific polarization of the immune system?
Response: As the reviewer commented, the term of immunostimulant is unclear. Like the following sentence, the term is simplified into to “exerts immunomodulatory effects.”
Do the authors mean IgA when they write type A antibody?
Response: We changed type A antibody to secretory immunoglobulin A.
Augmentation might not be the best word here.
Response: We changed it into just “activation.”
Is this the authors' hypothesis?
Response: NKA increase in the treatment group is our hypothesis. However, because the term of expectation may be overstated, we changed it to “PMBL administration may influence NKA.”
The authors' language belies bias toward an increase in NKA being beneficial. Perhaps a better word would be increases instead of improves.
Response: Thank you. We corrected it.
It remains unclear whether NK cell activity was measured as pg/mL interferon-gamma.
Response: We agree with the reviewer’s comment. NKA is surrogated by the IFN-γ stimulated with Promoca®.
The authors state that NKA <500 pg/mL is low according to the manufacturer. What is the lower limit of detection of this novel assay? This statement raises concern that the values obtained from sample measurements falls outside the standards range.
Response: The detection limit is 40 pg/mL. The cutoff of 500 pg/mL is defined low according to the manufacturer. However, a study suggested 438 pg/mL for gastric cancer, and our previous study showed time-dependent increase in NKA in the group with NKA <500 pg/mL. These points were additionally described in the method section (2.4. NKA measurement).
Why did the authors express general features as either parametric (i.e. mean +/- SD) or non-parametric (i.e. median + IQR)? Should it not be either one or the other, rather than both?
Response: The sample size of our study is large enough. In the revised manuscript, we showed the parametric parameters only.
If alpha = 0.05, then p ≤ 0.05 (rather than simply p < 0.05) is considered statistically significant.
Response: We corrected it.
It is unclear if the study design is powered for so many different p values. If there are 15 variables, some of which correlate with each other (e.g. BMI and health), one would expect some values to be "significant" by being ≤ 0.05. It is unclear if this is the case or if the study is powered sufficiently to account for all of these variables. Also, please indicate the primary end-point(s) and the secondary end-point(s).
Response: We added a sentence on the endpoint description. The most important point in our study is the change in NKA, that is the endpoint.
It is unclear why the authors split up the groups between whole and <500 pg/mL. If the patients in the control group were all healthy, why does it matter if they are split up into <500 vs whole? This is unclear to me why the authors would not also evaluate a ≥500 group if there is a <500 group.
Response: As mentioned above, in the subjects with NKA <500 pg/mL, time dependent increase in NKA was observed. We corrected the results into whole, NKA <500, and NKA ≥500 groups.
Why are these values being reported as parametric AND non-parametric? If the mean accurately represents the center of the sample distribution and the sample is large enough, parametric statistics should be used. Otherwise, non-parametric statistics should be used.
Response: Now, we clarified the endpoint. Since the endpoint is the change in NKA, we showed the parametric analyses only in the revised manuscript.
It would help to see the linear regression plots for these values. Standard errors ≥ ⅓ beta seem rather large.
Response: As we mentioned in the manuscript, age was not an expected variable. The other variables, such as neutrophil, lymphocyte, and NLR, were satisfied according to SE < ⅓ beta. We added the plots.
Why choose the medians instead of the means?
Response: Now we are using means.
This plot is measuring pg/mL interferon-gamma, not NKA. NKA is the interpretation of the plot. Also, thousands should be followed by a comma to separate them from hundreds (e.g. 3000 should be 3,000).
Response: We corrected them.
These values are down to femtogram ranges it appears. What is the limit of detection of this test?
Response: The detection limit is 40 pg/mL. We replaced the figure into a new one with mean and sem.
The syntax of this sentence leaves something to be desired.
Response: The sentence is corrected to “the PMBL treatment group with recurrent respiratory tract infection showed a greater increase in IFN-γ levels, a surrogate index for NKA.”
What is reasonable? Are the authors trying to say that it is reasonable to be concerned about this?
Response: We deleted “reasonable” in the sentence.
There is acquired and innate immunity. Inflammation and mucosal immunity are components of acquired and innate immunity, not alongside them. It is unclear what the authors are tyring to say here.
Response: We intended to say that PMBL is associated with acquired and innate immunity. In this part, we left only innate immunity.
Is this correlated with improved symptoms?
Response: Conclusion of this study mentioned that administration of PMBL® Tablet represents a safe and effective means for significantly reducing the rate of exacerbations in school-aged allergic asthmatic children. (Pediatr Allergy Immunol. 2018 Jun;29(4):394-401.)
SOME vaccines do; others do not.
Response: We agree with the reviewer’s comment. We corrected the sentence to avoid generalization.
Is it a vaccine or an immunostimulant? I do not think it is fair to call it a vaccine.
Response: Some developers insist to call PMBL as an oral vaccine. At present, it may be uncontroversial to call it an immune stimulant. We deleted “vaccine” in this sentence.
Reviewer 2 Report
The study was performed in a large group of individuals and both patients/controls enrollment and statistical anaylysis of obtained data is fine. However, my main concern is of NK cell activity analysis. The kit used for the study is not routinely used beside Korea, and no detailed protocol was either described in the manuscript, not is available online. What is more, NK cell activity cannot be limited to only IFNgamma production, since 1)in whole blood, used in the study, other cell types produce significant amounts of IFNg, i.e. monocytes, which can also be activated by PMBL, 2) NK cell activity is routinely measured by perforine and granzyme B production, degranulation or in direct cytotoxicity assay against K562 cells. Thus, the Authors cannot be sure that what they measured is really NK cell activity. Also, other data regarding NK cells, such as NK cell numbers, would improve the manuscript quality.
Author Response
The study was performed in a large group of individuals and both patients/controls enrollment and statistical anaylysis of obtained data is fine. However, my main concern is of NK cell activity analysis. The kit used for the study is not routinely used beside Korea, and no detailed protocol was either described in the manuscript, not is available online. What is more, NK cell activity cannot be limited to only IFNgamma production, since 1)in whole blood, used in the study, other cell types produce significant amounts of IFNg, i.e. monocytes, which can also be activated by PMBL, 2) NK cell activity is routinely measured by perforine and granzyme B production, degranulation or in direct cytotoxicity assay against K562 cells. Thus, the Authors cannot be sure that what they measured is really NK cell activity. Also, other data regarding NK cells, such as NK cell numbers, would improve the manuscript quality.
Response: We agree with the reviewer’s advice. As the reviewer commented, NKA measured by the method in this study is not widely used in the world. IFN-γ test stimulated by Promoca® is introduced recently and actively utilized in many studies. We added the measurement procedure in the method section.
We understand that IFN-γ can be secreted from other immune cells than NK cells. However, the NKA analysis method introduced in our study used activator Promoca® (PCT patent WO2012110878), which specifically stimulates NK cells to secrete IFN-γ. The references to support this specificity are also added in the method section.
NKA can be measured in various ways. As the reviewer commented, coincubation with K562 cells is an essential method to assess NKA. Unfortunately, we did not employ this method or count the numbers. Nonetheless, the method in this study has strengths for analyses on large samples. We added this limitation in the limitation part of Discussion.
Round 2
Reviewer 2 Report
The Authors have addressed and revised their manuscript according tomy comments.